Analysis of inferior nasal turbinate volume in subjects with nasal septum deviation: a retrospective cone beam tomography study

Shetty Shishir 1 sshetty@sharjah.ac.ae
Al-Bayatti Saad 1
http://orcid.org/0000-0001-7131-1752 Alam Mohammad Khursheed 2
http://orcid.org/0000-0002-7483-6594 Al-Rawi Natheer H. 1
Kamath Vinayak 3
Tippu Shoaib Rahman 4
Narasimhan Sangeetha 1
Al Kawas Sausan 1
Elsayed Walid 5
Rao Kumuda 6
Castelino Renita 6
1 Department of Oral and Craniofacial Health Sciences, University of Sharjah , Sharjah , United Arab Emirates
2 Preventive Dentistry Department, Jouf University , Sakaka, Al Jouf , Saudi Arabia
3 Department of Public Health Dentistry, Goa Dental College , Goa , India
4 Department of Diagnostic and Surgical Dental Sciences, Gulf Medical University , Ajman , United Arab Emirates
5 College of Dentistry, Gulf Medical University , Ajman , United Arab Emirates
6 Department of Oral Medicine and Radiology, Nitte (Deemed to be University) , Mangalore , India
Testarelli Luca
Electronic publication date: 2022 Sep 23
Publication date: 2022
Volume: 10
Electronic Location ID: e14032
Received 2022 Feb 1; Accepted 2022 Aug 17
Copyright: © 2022 Shetty et al.
Copyright year: 2022
Copyright holder: Shetty et al.
License: This is an open access article distributed under the terms of the Creative Commons Attribution License, which permits unrestricted use, distribution, reproduction and adaptation in any medium and for any purpose provided that it is properly attributed. For attribution, the original author(s), title, publication source (PeerJ) and either DOI or URL of the article must be cited.
License URL: https://creativecommons.org/licenses/by/4.0/

Keywords: Turbinates, Hypertrophy, Nasal septum, Volume, Cone beam tomography

Funding: University of Sharjah 2101100146 This study is funded by University of Sharjah, funding number V.C.R.G./R.438/2020 (Project number 2101100146). The funders had no role in study design, data collection and analysis, decision to publish, or preparation of the manuscript.

==============================
Background

The association of the linear dimensions of the inferior turbinate hypertrophy with nasal septal deviation has been studied recently. However, the volumetric dimensions provide a more accurate status of the turbinate hypertrophy compared to linear measurements. The aim of this study was to analyze the association of inferior nasal turbinate volume with the degree of nasal septal deviation (NSD).

Methods

A retrospective evaluation of the cone beam computed tomography (CBCT) scans of 412 patients was carried out to obtain 150 scans which were included in the study. The scans were categorized into three groups. Group 1 comprised of 50 scans of patients with no inferior turbinate hypertrophy (ITH) and no nasal septal deviation. Group 2 comprised of 50 scans of patients with ITH and no NSD; whereas Group 3 included 50 scans of patients with ITH and NSD. The total turbinate volume of inferior turbinates (bilateral) were determined by using Vesalius 3D software (PS-Medtech, Amsterdam, Netherlands).

Results

The intraclass correlation coefficient (ICC) between the volumetric estimations performed by the two radiologists was 0.82. There were no significant age and gender related changes in the total turbinate volume. Patients in Group 3 had significantly higher (p = 0.001) total turbinate volume compared to Group 2 and Group 1. There was a positive and significant correlation (r = 0.52, p = 0.002) between the degree of septal deviation and total turbinate volume. When the total turbinate volume of the patients with different types of septal deviation was compared in Group 3, a statistically significant difference (p = 0.001) was observed. Regression analysis revealed that the septal deviation angle (SDA) (p = 0.001) had a relationship with total turbinate volume. From the results of the study we can conclude that the total turbinate volume is higher in patients with nasal septal deviation. It can also be concluded that the septal deviation angle has a positive correlation with total turbinate volume. The data obtained from the study can be useful in post-surgical follow up and evaluation of patients with nasal septal deviation and hypertrophied inferior nasal turbinate.

Introduction

Prolonged obstruction of the nasal cavity is usually caused by nasal septal deviation (NSD) and inferior turbinate hypertrophy (ITH) (Abdullah & Singh, 2021). In most of the cases of ITH, medical line of treatment yields satisfactory results. However, in some instances when medical line of treatment does not provide satisfactory results surgical intervention is required (Aslan, 2013). Turbinate volume reduction is the main aim of all the protocols used for the treatment of ITH (Numminen et al., 2003). Although acoustic rhinometry (AR) is frequently used to study airway volume, accurate details of the especially regarding the posterior areas of the turbinate have been questionable (Numminen et al., 2003). Computed tomography (CT) provides the most accurate volume measurement of the paranasal structures and volume (Orhan, Aksoy & Oz, 2017). Though CT scans provide appropriate diagnostic quality, the radiation dose considerations are debatable (https://www.intechopen.com/chapters/55475#B7). Cone beam tomography (CBCT) has recently evolved as low radiation imaging modality for evaluating sinonasal structure (Orhan, Aksoy & Oz, 2017). Apart from lower radiation dose CBCT has higher image resolution, lower cost and provides thinner image slices compared to CT. In our previous work we have studied the association between NSD and inferior turbinate hypertrophy (ITH) using CBCT (Shetty et al., 2021). We found that the width of the inferior nasal turbinates had an association with the degree of nasal septal deviation. Based on the experience of our previous work we believe that the total turbinate volume of the inferior nasal turbinates may have an association with the degree of nasal septal deviation. We also wanted to determine whether there was association of the total turbinate volume with age and gender of the patients. To the best of our knowledge we did not find any studies investigating the total turbinate volume with the NSD. We believe that the results obtained from the volumetric analysis will aid in the diagnosis and management of cases with of NSD with ITH. With this background we conducted a study to determine the association between NSD and the volume of the inferior turbinates using CBCT.

Materials and Methods

A retrospective evaluation of CBCT scans of 412 patients who had attended University Dental Hospital Sharjah (UDHS) clinics for various dental treatments from January 2017 to December 2020 was carried out. Based on the eligibility criteria for the three groups to obtain 150 scans were selected for the study. Ethical approval for the study was obtained from the institutional ethical committee (Reference number: REC-21-01-10-01, University of Sharjah). Informed written consent was obtained from all patients involved in the study.

CBCT scans of male and female patients between 18 to 75 years of age were included in the study. The study data is available at figshare; DOI 10.6084/m9.figshare.19103570.

CBCT scans that were obtained using Galileos, Sirona CBCT Dental Systems (Bensheim Germany) x-ray machine (Field of View 15 × 24 cms and voxel size 0.25 mm), were used for this study. The machine was operated using SIDEXIS Operating system at 85 kVp and 7 mA. Assessment of CBCT was be performed directly on a 1920 × 1080 pixel and 23-inch DELL monitor screen.

Two dental radiologists with 10 years’ experience examined the CBCT scans. A third examiner with equivalent expertise was consulted in case of a disagreement between the two primary examiners. Sample size estimation was done using statistical Software G*Power 3.1. Based on the observation made from previous literature (El-Anwar et al., 2017; Orhan et al., 2014) and considering effect size of 0.26, 80% power and α error of 5%, a sample size of 49 was calculated which was rounded off to 50 per group.

The CBCT scans were screened between January 2021 and June 2021. The scans screened and included based on the grouping criteria until 50 from each group were obtained.

Group 1-50 patients with no ITH and no NSD.

Group 2-50 patients with ITH and no NSD.

Group 3-50 patients with ITH and NSD.

The criteria for determining ITH was based on the maximal width of inferior turbinate in the coronal CBCT section. A width of more than 10 mm was considered as ITH, based on the findings of published CT and CBCT studies (Jha, Ghimire & Shrestha, 2020; Shetty et al., 2021). Based on the maximal width of the right and left turbinates the scans were further subdivided as unilateral or bilateral ITH. CBCT scans in Group 1 and 2 had no evidence on NSD when viewed from the coronal section of CBCT at the point of crista galli. In CBCT scans of Group 3 the septal deviation angle was determined by the method used by Shetty et al. (2021) and Al-Rawi et al. (2019) (Fig. 1). The scans were further classified into mild (1° to 9°), moderate (10° to 15°), and severe (>15°). CBCT scans with artifacts, and incomplete anatomical coverage of the region of interest, scans of patients with mid facial trauma, orthognatic surgery and cleft palate were excluded from the study.

Figure 1 The radiographic landmarks used for determining the SDA (represented as angle ABC).

Point A represents the junction of the nasal septum with the floor of the nasal cavity. Point B represents the Crista Galli. The line BC represents a tangent drawn from point B and passing through the outermost part on the convexity of the deviated septum.

The total turbinate volume of inferior turbinates (bilateral) were determined by using Vesalius 3D software (PS-Medtech, Amsterdam, Netherlands). The volumetric analysis was performed by the two radiologists separately. To achieve uniformity in volumetric analysis a prior agreement was reached regarding the anterior, posterior and the lateral extent of the inferior nasal turbinates. The mean total turbinate volume was calculated. In the first step, CBCT scans were imported into the software. The inferior nasal turbinates were then segmented using surface in built extraction tools (scissors and eraser). After completing the segmentation of inferior nasal turbinates the volume was determined using the picking section of the software (Fig. 2).

Figure 2 The volume determination (yellow circle) of the segmented inferior nasal turbinates.

The data was statically analyzed using IBM SPSS statistics (Version 22, IBM Corp, Armonk. NY, USA). The total turbinate total turbinate volume of the patients among the three groups was compared using the ANOVA and Tukey Post Hoc Test. The total turbinate volume of the patients with different types of septal deviation in Group 3 was compared using ANOVA and Tukey Post Hoc Test. The correlation between total turbinate volume and septal deviation angle in Group 3 was determined using Pearson’s Correlation Test. The total turbinate volume between the male and female patients in study groups was compared using Independent sample t test. Regression analysis and dot plots were used to describe the correlation/difference between turbinate volume and other continuous/categorical factors.

Results

The intraclass correlation coefficient (ICC) between the volumetric estimations performed by the two radiologists was 0.82. Each examiner re-evaluated 5% of the scans from the total samples after a gap of 15 days to determine the intra-examiner reliability (ICC = 0.93).

This value indicates a good reliability of the volumetric estimation technique used in the study.

When the total turbinate total turbinate volume of the patients among the three groups was compared, there was statistically significant difference (p = 0.001) (Tables 1 and 2). Patients in Group 3 had significantly higher total turbinate volume when compared to Group 1 and Group 2. Patients in Group 1 had lowest total turbinate volume among the groups.

Table 1 Comparison of total turbinate volume of the study subjects among the study group.

Study groups	N	Mean	SD	Min	Max	ANOVA	
F	p-value	
Group 1	50	3,228.44	314.04	2,699.78	3,728.56	759.10	0.001*	
Group 2	50	4,354.58	289.15	3,722.01	4,833.40	
Group 3	50	5,347.12	199.54	4,844.66	5,598.09	
Notes:

* p < 0.05 statistically significant.

p > 0.05 non-significant, NS.

Table 2 Pairwise comparison of total turbinate volume between the study groups.

(I) Group	(J) Group	Mean difference (I-J)	Std. error	p-value	95% Confidence Interval	
Lower bound	Upper bound	
Group 1	Group 2	−1,126.14	54.41	0.001*	−1,254.97	−997.31	
Group 3	−2,118.68	54.41	0.001*	−2,247.51	−1,989.85	
Group 2	Group 3	−9,92.54	54.41	0.001*	−1,121.37	−863.71	
Notes:

Tukey Post Hoc Test.

* p < 0.05 statistically significant.

p > 0.05 non-significant, NS.

On comparison of the total turbinate volume between patients with unilateral and bilateral hypertrophy in Group 2, there was no statistically significant difference (p = 0.99). However, there was a significant difference (p = 0.01) of the total turbinate volumes between patients with unilateral and bilateral turbinate hypertrophy in Group 3 (Table 3). This suggested that in the presence of NSD there is a significant increase in the total turbinate volume in patients with unilateral turbinate hypertrophy.

Table 3 Comparison of total turbinate volume between the type of hypertrophy in study groups.

Group	Type of hypertrophy	N	Mean	SD	Mean difference	95% Confidence Interval of the difference	t	df	p-value	
Lower	Upper	
Group 2	Unilateral	15	4,354.96	296.64	0.55	−180.72	181.83	0.006	48	0.99 (NS)	
bilateral	35	4,354.41	290.27	
Group 3	Unilateral	38	5,427.79	128.37	336.14	243.94	428.34	7.33	48	0.01*	
bilateral	12	5,091.65	168.09	
Notes:

Independent sample t test.

* p < 0.05 statistically significant.

p > 0.05 non-significant, NS.

When the total turbinate volume of the patients with different types of septal deviation was compared in Group 3, a statistically significant difference (p = 0.001) was observed (Table 4). However, during pairwise comparison the difference in total turbinate volume between mild and moderate type of septal deviation was not significant (p = 1.0) (Table 5).

Table 4 Comparison of total turbinate volume between septal deviation type in study group 3.

Septal deviation type	N	Mean	SD	Min	Max	ANOVA	
F	p-value	
Mild	16	5,284.01	204.53	4,844.66	5,533.08	7.50	0.001*	
Moderate	19	5,283.12	175.57	4,933.11	5,498.70	
Severe	15	5,495.49	145.62	4,987.09	5,598.09	
Notes:

* p < 0.05 statistically significant.

p > 0.05 non-significant, NS.

Table 5 Pairwise comparison of total turbinate volume between septal deviation type in study group 3.

(I) Septal deviation type	(J) Septal deviation type	Mean difference (I-J)	Std. error	p-value	95% Confidence Interval	
Lower bound	Upper bound	
Mild	Moderate	0.89	60.20	1.00 (NS)	−144.79	146.57	
Severe	−211.48	63.76	0.005*	−365.78	−57.17	
Moderate	Severe	−212.37	61.27	0.003*	−360.66	−64.07	
Notes:

Tukey Post Hoc Test.

* p < 0.05 statistically significant.

p > 0.05 non-significant, NS.

Correlation of the total turbinate volume in patients of study group 3 with septal deviation angle revealed a moderate statistically significant correlation (r = 0.52, p = 0.001) (Table 6). This finding suggests that the total turbinate volume tends to increase as the degree of nasal septal deviation increases.

Table 6 Correlation between total turbinate volume and septal deviation angle in group 3.

Group	Group 3	
Septal deviation angle	r	0.52	
p-value	0.002*	
Notes:

Pearsons Correlation Test.

* p < 0.05 statistically significant.

p > 0.05 non-significant, NS.

There was no significant correlation between total turbinate volume and age of the patients in the study groups (Table 7). There was no statistically significant difference in the total turbinate volume between the male and female patients of the study groups (Table 8).

Table 7 Correlation between total turbinate volume and age of the subjects in the study groups.

Group	Group 1	Group 2	Group 3	
Age	r	0.25	0.33	0.31	
p-value	0.08 (NS)	0.74 (NS)	0.86 (NS)	
Notes:

Pearsons Correlation Test.

* p < 0.05 statistically significant.

p > 0.05 non-significant, NS.

Table 8 Comparison of total turbinate volume between the male and female subjects in the study groups.

Group	Gender	N	Mean	SD	Mean difference	95% Confidence Interval of the difference	t	df	p-value	
Lower	Upper	
Group 1	Male	32	3,242.94	312.48	40.29	−147.31	227.89	0.43	48	0.67 (NS)	
Female	18	3,202.65	324.21	
Group 2	Male	33	4,343.92	302.10	−31.35	−206.47	143.78	−0.36	48	0.72 (NS)	
Female	17	4,375.27	269.87	
Group 3	Male	37	5,364.62	195.31	67.31	−61.92	196.53	1.05	48	0.3 (NS)	
Female	13	5,297.31	210.96	
Notes:

Independent sample t test.

* p < 0.05 statistically significant.

p > 0.05 non-significant, NS.

Regression analysis of the parameters in group 3 revealed that septal deviation type (p = 0.001), septal deviation angle (p = 0.001) and type of hypertrophy (p = 0.001), had significant relationship with the total turbinate volume. Whereas no significant change in the total turbinate volume was observed when the gender (p = 0.68) and age (p = 0.53) of the patients was considered (Table 9 and Fig. 3).

Table 9 Linear regression to predict total turbinate volume based on study variables.

	Unstandardized coefficients	Standardized coefficients	t	p-value	95.0% Confidence Interval for B	
B	Std. error	Beta	Lower bound	Upper bound	
(Constant)	3,385.34	184.16		18.38	0.001*	3,021.36	3,749.32	
Gender	−33.73	81.64	−0.02	−0.41	0.68 (NS)	−195.08	127.62	
Age	1.64	2.58	0.03	0.64	0.53 (NS)	−3.46	6.74	
Septal deviation angle	91.73	6.26	0.63	14.65	0.001*	79.36	104.10	
Type of hypertrophy	563.0	49.57	0.50	11.36	0.001*	465.01	660.98	
Notes:

F (5,145) = 106.97, p < 0.001, R2 = 0.75.

* p < 0.05 statistically significant.

p > 0.05 non significant, NS.

Figure 3 Dot plots showing the correlation/difference between turbinate volume and other factors (A. Gender, B. Study groups, C. Septal deviation angle and D. Type of hypertrophy).

Discussion

Chronic nasal obstruction can be caused by several factors including NSD and mucosal turbinate hypertrophy of the nasal turbinates (Kumar, Anand & Pal, 2017). In order to reduce nasal obstruction and improve air passage into the nasal valve, reduction in the volume of ITH and correction of NSD should be addressed simultaneously (Illum, 1997). Earlier studies have investigated the linear dimensions of ITH and correlated them with SDA (Chiesa Estomba et al., 2015). Recent studies have revealed that volumetric measurements are more accurate in demonstrating the extent of the pathology compared to the linear measurements (Clarke et al., 2012; Pérez Sayáns et al., 2020). In the present study we have determined the volume of ITH and associated it with NSD. Vesalius SD software was used for the purpose of segmentation and volume determination of the ITH in the present study. Recent studies have demonstrated that Vesalius 3D provides precise volumetric analysis of the anatomic structures (Almgoter & Al-Dahan, 2020).

In the present study there was no significant change in the total turbinate volume of patients when the age was considered. Similar age related findings were observed in a computed tomography CT based study (Uzun et al., 2004). However, the turbinates were evaluated on the basis of linear dimensions observed on the CT scans in the later study (Uzun et al., 2004).

In the present study there was no significant difference in the gender distribution of the total turbinate volume. In a recently published CT based study conducted to evaluate three dimensional (3D) polymorphism of the inferior turbinates, no gender dimorphism was observed (de Bonnecaze et al., 2018).

In our study the total turbinate volumes of the three groups were compared. The categorization of scans in three groups was intended to assess the change in the total turbinate volume in the presence of NSD and ITH. The total turbinate volume of the group having NSD with ITH had significantly higher turbinate volumes compared to the other groups (without NSD and ITH).

The hypertrophy of inferior turbinates occurs as a compensatory reaction in individuals with NSD. It has been found that the hypertrophy is not just caused by mucosal hypertrophy, but also by hypertrophy of the turbinate bone (Orhan et al., 2014). Though the soft tissue and bone component contribute to the ITH, the bony component is responsible for the major contribution (Berger et al., 2000). However, it is important to note that when the bony and soft tissue components are individually correlated with the degree of nasal septal deviation it was not significant (Akoğlu et al., 2007). Therefore, it can be inferred that the volumetric analysis has a more significant correlation when compared to linear measurements of the bony or soft tissue components of the ITH.

In the present study there was no significant difference in the total turbinate volume in mild and moderate type of NSD. However, the total turbinate volume difference was highly significant between mild and severe type of NSD. Similarly, the total turbinate volume was significantly higher in severe type of NSD when compared to mild type of NSD. In earlier studies conducted by Grymer et al. (1989, 1991), Grymer, Illum & Hilberg (1993) compensatory inferior turbinate hypertrophy was observed more commonly in the septal deviation of moderate or severe degree of NSD.

From the results of our study it can be inferred that patients with NSD have significantly higher total turbinate volume when the hypertrophy is unilateral in nature.

The increase in total volume in unilateral type of hypertrophy is a compensatory mechanism to reduce the adverse effects of one sided breathing resulting from NSD (Sharma, 2016). The adverse effects commonly associated with NSD include dryness of the nasal passage, alteration of air filtration, and mucociliary flow (Teixeira et al., 2016).

In the present study regression analysis revealed a positive relationship between total turbinate volume and SDA. In a study by Tomblinson et al. (2016) regression analysis revealed that the ITH thickness had positive relationship with severity of NSD.

The results of the present study demonstrate the effectiveness of three dimensional volumetric analyses of turbinates in CBCT scans. Recently a study was conducted by Valtonen et al. (2021) to evaluate the volume of ITH pre operatively and post-operatively. Results of the study revealed that CBCT scans offer more complete and precise information of the turbinates. The results of their study also revealed that three dimensional analyses of the ITH provides more accurate dimensions compared to acoustic rhinometry (Valtonen et al., 2021). Studies have revealed that the volumetric measurements in the middle and posterior parts of the nasal passage are not accurately measured by acoustic rhinometry because of the sound loss through ostia. This often causes overestimation of airspace cross section and volume (Cankurtaran et al., 2007; Hilberg et al., 1998; Terheyden et al., 2000). The use of CBCT or CT based volumetric analysis software can be used to overcome this problem. In the present study Vesalius 3D was used to perform volumetric analysis of the ITH, whereas in the study by Valtonen et al. (2021) OnDemand3D™ software was used to perform three dimensional volumetric measurements.

One of the major limitations of our study is the time required for the volumetric analysis of ITH. The semi-automated segmentation software considerably reduces segmentation time and improves accuracy compared to manual segmentation (McGrath et al., 2020). However, the segmentation procedure still takes considerable amount of time. This can be overcome in the future by using deep learning based fully automated segmentation software which can provide faster results with minimal intra examiner and inter examiner variabilities (Vaidyanathan et al., 2021).

Conclusions

The results of the study reveal that the total turbinate volume is higher in patients with nasal septal deviation. It can also be concluded that the septal deviation angle has a positive correlation with total turbinate volume. In the future, volumetric analysis of turbinates can be used in the pre-treatment evaluation and post treatment follow-up of patients with inferior turbinate hypertrophy with nasal septal deviation.

Additional Information and Declarations

Competing Interests

Author Contributions

Human Ethics

Data Availability

Mohammad Khursheed Alam is an Academic Editor for PeerJ.

Shishir Shetty conceived and designed the experiments, performed the experiments, analyzed the data, prepared figures and/or tables, authored or reviewed drafts of the article, and approved the final draft.

Saad Al-Bayatti conceived and designed the experiments, performed the experiments, analyzed the data, prepared figures and/or tables, and approved the final draft.

Mohammad Khursheed Alam conceived and designed the experiments, performed the experiments, prepared figures and/or tables, and approved the final draft.

Natheer H. Al-Rawi conceived and designed the experiments, analyzed the data, prepared figures and/or tables, authored or reviewed drafts of the article, and approved the final draft.

Vinayak Kamath conceived and designed the experiments, performed the experiments, analyzed the data, prepared figures and/or tables, authored or reviewed drafts of the article, and approved the final draft.

Shoaib Rahman Tippu conceived and designed the experiments, analyzed the data, prepared figures and/or tables, and approved the final draft.

Sangeetha Narasimhan conceived and designed the experiments, prepared figures and/or tables, authored or reviewed drafts of the article, and approved the final draft.

Sausan Al Kawas conceived and designed the experiments, analyzed the data, prepared figures and/or tables, authored or reviewed drafts of the article, and approved the final draft.

Walid Elsayed conceived and designed the experiments, authored or reviewed drafts of the article, and approved the final draft.

Kumuda Rao conceived and designed the experiments, prepared figures and/or tables, authored or reviewed drafts of the article, and approved the final draft.

Renita Castelino conceived and designed the experiments, prepared figures and/or tables, authored or reviewed drafts of the article, and approved the final draft.

The following information was supplied relating to ethical approvals (i.e., approving body and any reference numbers):

Ethical approval for the study was obtained from the University of Sharjah institutional ethical committee (Reference number: REC-21-01-10-01, University of Sharjah).

The following information was supplied regarding data availability:

The data is available at Figshare: Shetty, Shishir (2021): Analysis of inferior nasal turbinate volume in subjects with nasal septum deviation: a retrospective cone beam tomography study. figshare. Dataset. https://doi.org/10.6084/m9.figshare.17067647.v10.

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
