# Peer review of "Analysis of inferior nasal turbinate volume in subjects with nasal septum deviation: a retrospective cone beam tomography study"

_PeerJ, doi:10.7717/peerj.14032_

## Round 0.1 · original submission · Major Revisions

Thank you for your submission. Please find the comments below.

Reviewer 1 ·

Basic reporting

I would suggest further modification before any revision.

Experimental design

I would suggest further modification before any revision.

Validity of the findings

I would suggest further modification before any revision.

Additional comments

The topic is relevant. However, the name of subjects is displayed in https://dx.doi.org/10.6084/m9.figshare.19103570. Please, address it. The ethical issue here is too problematic. I would suggest the authors to contact their IRB to express this aspect before any attempt to publish the material. The exposure of names is not a regular practice in well-conducted science.

Reviewer 2 ·

Basic reporting

a. The study is written in a professional scientific manner with sufficient literature references. However, the clarity of the manuscript needs to be improved with some reorganizing of content and rephrasing.
i. In Abstract: Abstract is difficult to follow. The results in abstract not explained in methods. Suggest picking the two most important results and clearly state the methods used to arrive at the result. Line 35: need to state the group, line 38 and 39: need to rephrase for clarity, line 4: difference in total turbinate volume compared to? Line 42: type of septal deviation not defined. Suggest omitting as does not add to the conclusion in the abstract? SDA: this needs to be spelled out at first mention then. Suggest to omit the word significant if need to reduce word count as P value of <0.05 is universally understood.
ii. Introduction: line 67: Is it total volume or only volume of the hypertrophied inferior turbinate.
iii. Figure 2 is not referenced in the text
iv. In discussion: please justify and explain how dividing the scans into three groups further adds to answer the objectives. Please discuss the result which compares these three groups in discussion.
v. Suggest to layout the results in term of clinical importance. Age and gender relation to inferior turbinate volume are secondary objectives and should be presented later. The comparison of Inferior turbinate volume between three groups should be mentioned first and be well discussed along with its clinical relevance.

Experimental design

b. The research question (aim) is not well defined. There is insufficient justification for this study. This paper needs to further emphasize the importance of studying the volume of Inferior turbinate in relation to NSD. Line 68: suggest rephrase. Association between NSD AND total volume inferior turbinate? The hypothesis should also be expanded to include the secondary objectives. Ie. the association of inferior turbinate volume and age, gender, and type of inferior turbinate hypertrophy. relevant & meaningful. Rigorous investigation performed to a high technical & ethical standard.
c. There is lack of novelty n this study. The authors needs to state the importance of volumetric analysis of inferior turbinate and how this will help advance the management of nasal obstruction.
d. The Methods need to be further described.
a. Please state how the Inferior turbinate hypertrophy was differentiated from no hypertrophy based on the CBCT in order to classify the scans into the three group. Please cite the validated definition of turbinate hypertrophy on CT scan.
b. How do authors define no septal deviation. In classification of NSD mild is <9 degree. Please clarify what is accepted as no deviation. If patients have 2 degree deviation, is this still defined as NSD? NSD is very common and most patients will exhibit some form of NSD. Lee at al, 2021 (Lee, Joshua A., et al. "Radiological Assessment of High Anterior Septal Deviation and Its Impact on Sinus Access." The Laryngoscope (2021) reported the septal deviation angle between those who received septoplasty and those who do not. The reported values for SDA (11.1° ± 4.3 vs. 7.3° ± 3.4; d = 1.00 [95% CI 0.58–1.40]). Therefore even patients who do not need septoplasty have some form of septal deviation. Based on the raw data, no septal deviation group has zero degree SDA. This needs further explanation.
c. How was mild DNS but no ITH dealt with?
d. Please state how 150 CBCTs were selected from the scan archives. Was there a specified period of time? Were the scans screened and consecutively included based on the grouping criteria until 50 from each group were obtained? What were the inclusion and exclusion criteria.
e. Unilateral and bilateral ITH stated in result need to be defined in methods for clarity.

Validity of the findings

The validity is becomes questionable as the selection criteria and definition of inferior turbinate hypertrophy and DNS is not defined.
The authors need to state how this can be translated clinically. CBCT is not routine for the evaluation of nose block, therefore, please discuss how the results from this CBCT study can be translated into clinical practice.

Additional comments

In general, this study lack novelty. The study is mainly repetitive with similar previous findings. The selection process needs to be described. The present or absent of ITH, as well as DNS needs to be well defined and logical.

Reviewer 3 ·

Basic reporting

The study examined the correlation between severity of ITH and three-dimensional turbinate volume estimated from CBCT scans, suggesting use of CBCT could potentially promote ITH treatment. The paper is writing in professional English language, and it’s easy to follow and interesting to read. Sufficient literatures referenced and comprehensive background provided.

Experimental design

The topic of this study is within Aims and Scope of the journal. Authors proposed meaningful aims/questions with clear rationales. However, the results will be more reliable if some holes of the statistical methods will be revised.

Validity of the findings

Results were well stated in a logical way and the research question was answered. This study is highly worthy to be added into current literature upon revising statistical parts.

Additional comments

1) It is unclear to me that how results on Table 1 and Table 3 help the overall conclusion of the study. The conclusion does not seem to be diminished by removal of these tables.
2) It is possible that the linear correlation detected by the regression model only exists within certain range of turbinate volume and correlation pattern might vary as turbinate volume changes. I would like to suggest dot plots to visually show the correlation/difference between turbinate volume and other continuous/categorical factors in addition of the regression model
3) ANOVA assumes that data distributes normally within each group and variance across groups are not significantly different. Have these assumptions been checked? If these assumptions are not met, Kruskal-Wallis test could be an alternative for ANOVA.
4) There should be a paragraph describing statistical methods used in the study at the end of “Materials & Methods” section, with literature cited if applicable.
5) Since the septal deviation type (<9 for mild, 9-15 for moderate, >15 for severe) was developed from a septal deviation angle, it will potentially cause multicollinearity when fitting both variables into regression model and result in less reliable statistical inferences. I would like to suggest dropping the septal deviation type and only using the septal deviation angle since it carries granulated information which will make more precise prediction.
6) It is a little bit confusing that some parts used ANOVA followed by pairwise comparison while other results used t-tests for each pair of groups.
7) Is there any previous studies showed the accuracy of estimated turbinate volume from CBCT scans?
8) Line 40: it should be “There was a positive...”
9) Line 94: Reference [5], change format to be consistent with other references
10) Statistical method descriptions should be moved to the statistical methods paragraph(s)
a. Line 109-110: “When the total turbinate … using the ANOVA and Tukey Post Hoc Test”
b. Line 118-119: “When the total turbinate … using ANOVA”
11) Line 116: use “Table 7” instead of “table 7” to be consistent with the rest of manuscript, and so dose for “(p=0.69)” on line 127
12) Are there any other factors influencing turbinate volume but were not able to include in the study?

---

## Round 0.2 · Minor Revisions

Thank you for your submission. Please find the comments attached and do the necessary revisions.

Reviewer 1 ·

Basic reporting

No comment

Experimental design

no comment

Validity of the findings

no comment

Additional comments

I would suggest the authors to avoid use “subjects”. A better nomenclature should be adopted.

The abstract is not complete. Some parameters are used in results but not appear in methods.


Add all meaning for abbreviation the first time it appears in text. For example, CBCT in abstract. SDA? Moreover, “nasal septal deviation” is used for the first time in background (abstract) as with no abbreviation...

How a retrospective study could easily reach a similar sample for 3 groups? Do all potential patients were screened for inclusion/exclusion criteria? The inclusion of patients stopped when 50 individuals were included per group? How about the total sample size? Total cohort?

It is not reasonable to include 150 patients and reach 3 groups with 50 patients per group. How about the total cohort before application of inclusion/exclusion criteria?

Why “total turbinate volume of inferior turbinates (bilateral)” was used as an outcome paramenter? Why not apply a difference between larger and smaller sizes?

How about the volume of the inferior turbinate in the side with deviation versus the volume of the inferior turbinate in the side with no deviation?

The side of deviation versus the turbinate volume of inferior turbinates should also be considered.


“We also wanted to determine whether there was association of the total turbinate volume with age and gender of the study subjects.” Why you tested this hypothesis?

“With this background we conducted a study to determine the association between NSD on the volume of ITH using CBCT.” Is it correct? Maybe: the association between NSD and the volume of the inferior turbinates using CBCT? Or, the impact of NSD on the volume of the inferior turbinates...


This is a retrospective study design. How the sample calculation was calculated? Which type of method was used for sample calculation?

How many patients were screened until 150 individuals were included?


“method used by Jha et al., 2020 and Shetty et al., 2021”. Please, redo this writing style. I would suggest a more scientifically pleasant format.


How about the intra-evaluator and inter-evaluator reliabilities? How about some attempt to blind the evaluators?

How about the normality distribution of data?


The limitations of this study should be better detailed at the end of discussion section.

Reviewer 2 ·

Basic reporting

no comment

Experimental design

no comment

Validity of the findings

no comment

Additional comments

The authors have addressed my concers sufficiently. Nothing futher to add.

Reviewer 3 ·

Basic reporting

The authors have addressed all my comments. I have no concerns about the content. I would suggest the authors to do editorial proofreading since there are some typos and excessive words in sentences.

Experimental design

The authors have addressed all my comments. I have no concerns about the content. I would suggest the authors to do editorial proofreading since there are some typos and excessive words in sentences.

Validity of the findings

The authors have addressed all my comments. I have no concerns about the content. I would suggest the authors to do editorial proofreading since there are some typos and excessive words in sentences.

Additional comments

Some typos (not a comprehensive list):
1. Line 153, "(P=1.0) Table 5)"?
2. Line 163, "hand significance" -> "had significance"
3. Line 164, "(Table 9.(Figure 3)"?
4. choose between "Group 1" and "group 1" and make it consistent through the manuscript
5. Line 151, "Group 3a"?

---

## Round 0.3 · accepted · Accept

Dear Authors,

You made a great work! Your manuscript has been accepted!

Reviewer 2 ·

Basic reporting

No further comments

Experimental design

No further comments

Validity of the findings

No further comments

Additional comments

No further comments

Reviewer 3 ·

Basic reporting

All my comments have been addressed. I don't have further concerns about the manuscript.

Experimental design

All my comments have been addressed. I don't have further concerns about the manuscript.

Validity of the findings

All my comments have been addressed. I don't have further concerns about the manuscript.

Additional comments

All my comments have been addressed. I don't have further concerns about the manuscript.